# A Case–Control Study Examining the Association of Fiber, Fruit, and Vegetable Intake and the Risk of Colorectal Cancer in a Palestinian Population

**DOI:** 10.3390/ijerph19127181

**Published:** 2022-06-11

**Authors:** Hania M. Taha, Alexander N. Slade, Betty Schwartz, Anna E. Arthur

**Affiliations:** 1Department of Food Science and Human Nutrition, University of Illinois at Urbana-Champaign, Urbana, IL 61801, USA; haniat2@illinois.edu; 2Augusta Victoria Hospital, The Lutheran World Federation, East Jerusalem 91191, Palestine; 3Department of Radiation Oncology, Stony Brook University School of Medicine, Stony Brook, NY 11794, USA; alexander.slade@stonybrookmedicine.edu; 4Robert H. Smith Faculty of Agriculture, Food, and Environment, Hebrew University of Jerusalem, Rehovot 761001, Israel; betty.schwartz@mail.huji.ac.il; 5Department of Dietetics and Nutrition, University of Kansas Medical Center, Kansas City, KS 66160, USA

**Keywords:** colorectal cancer, dietary fiber, fruits and vegetables, Palestinian population, dietary prevention, eating habits, case–control study

## Abstract

While there is an association between Western diets and the incidence of colorectal cancer (CRC), this dietary association has remained unexplored in Palestine. The aim of this study was to examine how fiber and fruit and vegetable (FV) intakes are associated with CRC risk among Palestinian adults. We recruited 528 Palestinians between 2014 and 2016. We identified 118 patients who received CRC treatment at Augusta Victoria Hospital, East Jerusalem. We additionally identified 410 controls who consisted of community-based Palestinians without cancer. All participants completed a survey on demographics and a validated dietary intake food screener. Multivariable logistic regression models tested associations between fiber and FV intakes (categorized into quartiles) with CRC risk. After adjusting for significant covariates (age, sex, education, physical activity, smoking status, BMI, IBD, and family history of CRC), as fibers increased across increasing quartiles, the CRC risk significantly decreased (OR = 0.36, 95% CI: 0.15–0.86, *p*-trend = 0.02). After adjusting for age and sex, as FV intake increased, the CRC risk significantly decreased (OR = 0.34, 95% CI: 0.15–0.75, *p*-trend = 0.009). Consumption of fiber-rich foods was inversely associated with CRC risk. Understanding this relationship among Palestinians is essential in order to develop targeted, culturally relevant strategies that may potentially alleviate the burden of CRC.

## 1. Introduction

According to GLOBOCAN 2020, colorectal cancer (CRC) is the third most diagnosed type of cancer worldwide after lung and breast cancers, representing nearly 10% of all cancer diagnoses [1]. It is also the second leading cause of cancer-related death after lung cancer, representing 9.4% of all cancer-related deaths worldwide [1]. Although CRC incidence is most common in high-income countries [2], CRC is expected to increase significantly in developing countries by 2030 [3], including in the Middle East [4], due to environmental and lifestyle changes.

Based on statistics available from 2020, which were limited in part due to the political and geographic separation of the West Bank, Gaza Strip, and East Jerusalem, CRC was the second most commonly diagnosed cancer among Palestinians after breast cancer (and the first among men) with a prevalence rate of 13.1% and an incidence rate of 15.2%, mostly among adults older than 45 years [5]. CRC was also the second leading cause of cancer death in 2020 (13.9%), while there was a 2.1% increase in CRC mortality between the years 2017–2020 [5,6,7]. Furthermore, the documented incidence of CRC cases among Palestinians nearly doubled from 2013 to 2015 [8]. It is hypothesized that this rapid increase in CRC incidence was due, in part, to the absence of an organized program for CRC screening, where CRC is being diagnosed at later stages after symptoms are developed, [9] but also in part due to the epidemiologic transition characterized by rapid urbanization, environmental changes, and lifestyle changes, including a nutritional transition from traditional dietary patterns to more Westernized dietary habits, more sedentary lifestyles, and obesity [10,11]. The cancer burden among Palestinians is expected to increase even further in the coming years, exacerbated by limited access to prevention and early detection and underdeveloped cancer control strategies [9,12].

While there are strong, non-modifiable risk factors associated with CRC such as age, sex, personal history of inflammatory bowel disease (IBD), a family history of CRC, personal history of CRC polyps or CRC, and an inherited syndrome, there are also several modifiable risk factors that have been identified. These include obesity, lack of regular physical activity (PA), smoking, alcohol drinking, and eating patterns high in red or processed meat and low in fruits, vegetables, and fibers [10]. Among Palestinians, smoking is prevalent [13], sedentary lifestyles are common [13,14], and rates of overweight and obesity are increasing [13]. Diet is transitioning from being mainly traditional and Mediterranean (including legumes and whole wheat bread) to becoming more carbohydrate-rich, where there is an increased intake of refined foods, soft drinks, and sweets [15]. However, the association between CRC and the intake of fibers, fruits and vegetables (FV) on CRC risk has not yet been examined in the Palestinian territories. The purpose of this study is to examine the associations between intakes of fibers and FV and the risk of CRC among a sample of Palestinians adults. Understanding the dietary risk factors among Palestinians can help in setting population-specific guidelines for the prevention of CRC. 

## 2. Materials and Methods

### 2.1. Study Design and Population

The present research is a case–control study that includes 528 Palestinians > 18 years old (men and women). A convenient sample of 118 cases (patients with CRC treated for cancer at the Augusta Victoria Hospital (AVH) in East Jerusalem between 2014 and 2016) and 410 controls (individuals from the community without CRC) were recruited over ~two years (2014–2016). Most Palestinian CRC patients receive their CRC treatment at AVH; therefore, we consider this sample size to be representative of the Palestinian population who are diagnosed with CRC. Ethical approval was obtained from the Committee for the Use of Human Subjects in Research at the affiliated Hebrew University of Jerusalem. Inclusion criteria were as follows: age > 18; living in East Jerusalem, the West Bank, or Gaza strip; no history of dementia; ability to understand and sign a consent form. Further criteria for selecting cases included: diagnosed with cancer of either the colon or rectum with any type and stage and receiving treatment at AVH. The inclusion criteria for selecting controls was having no previous cancer diagnosis. Healthy controls were selected from the community, including the patients’ companions and people with diabetes who received nutritional counseling at AVH. We also included AVH employees and other healthy individuals from the community. Controls were given the surveys either in hard copies to fill out or were sent the survey via SurveyMonkey.com.

### 2.2. Data Collection

All participants submitted a signed consent form and completed a survey that included two sections; information about health-related factors and an online short-validated fruit, vegetable, and fiber screener to assess the intakes [16,17]. Data were collected from cases and controls through face-to-face interviews conducted by a trained dietitian or an online Qualtrics survey to reach a higher number of participants. Health-related factors on the survey included questions to collect data on sociodemographic characteristics and lifestyle habits, including age, sex, years of education, family history of CRC, personal history of IBD, usual body weight and height to calculate body mass index (BMI), PA, smoking status, and history of type II diabetes (DMII). PA was assessed by asking if the participant exercises for about 30 min for at least 3–4 days/week and what PA is involved in their work (sitting, standing/walking, physical effort, housework, or if they are unemployed). Alcohol drinking status was not assessed due to cultural sensitivity. Intake of red and processed meats was not assessed due to a lack of a validated screener that complies with the Palestinian intakes. The food screener was adapted and translated to Arabic from Nutrition Quest [17], due to its simplicity and comprehensiveness. It was composed of seven items (10 questions) asking about the frequency of intake of fibers and FV in the past year, response categories: <once/week, about once/week, 2–3 times/week, 4–6 times/week, every day, and ≥2 times/day, and it was designed to capture the intake of FV in servings/day and fibers in grams/day. The 10 questions were translated into Arabic by qualified research staff; then back-translated to English to ensure the translation’s accuracy. This forward–backward translation method has been used by the World Health Organization [18]. Visual aids of food items were provided if participants were not familiar with the name of food items. The screener was previously validated against a full length-validated food frequency questionnaire (FFQ) of 100 items calculated using USDA Food Pyramid definitions of servings and shown to be effective in identifying persons with both low FV (<5 servings per day) and fiber intakes (<25 g per day) [16]. Although originally designed for the general American population, investigators were familiar with foods consumed in the Palestinian community and deemed the food screener to be culturally appropriate to use in this study population. Screener responses were entered into and automatically analyzed for food servings via the NutritionQuest website [17]. For FV, food screener responses were categorized as follows: very low (0–2 servings/day), low (around 3 servings/day), medium (4–5 servings/day), or adequate (>5 servings/day). Fiber intake was estimated in total daily grams from dietary fibers and FV intakes.

### 2.3. Statistical Analyses

For statistical analysis, FV intakes were modeled categorically based on the results from the screener: 0–2 serving/day, 3 serving/day, 4–5 servings/day, and >5 serving/day. Fiber intake was categorized into quartiles (Q) based on the total sample intake. Multivariable logistic regression models were used to test the associations between fibers and FV intakes and CRC risk. Associations were examined in three separate multivariable logistic regression models. Model 1 (basic model) included demographic variables only (age and sex). Model 2 included demographic and clinical variables (Model 1 + IBD and family history of CRC). Model 3 included demographic, behavioral, and clinical variables (Model 1 + Model 2 + PA, smoking status, educational level, and BMI). Age was modeled as a continuous variable, and all other covariates were modeled categorically. PA was categorized into poor (exercising <150 min/week and work involves sitting only), moderate (exercising <150 min/week and work involves standing/walking/definite physical effort), and good (exercising for >150 min/week). Smoking status was categorized into never, former, and current. Educational level was categorized into ≤14 years of education, 15–16 years of education, and ≥17 years of education. BMI (kg/m^2^) was categorized into underweight (BMI < 18.5), normal (18.5 < BMI < 24.9), overweight (25 < BMI < 29.9), and obese (BMI ≥ 30). The lowest category of intake for fiber (Q1) and FV (0–2 servings/day) was set as the referent. Odds ratios (OR), 95% confidence intervals (CI), *p*-values < 0.05 were considered statistically significant, and tests for linear trends across increasing quartiles of intake were performed by setting each individual’s nutrient value to the median for that quartile and treating it as a continuous variable in logistic regression models. All statistical analyses were performed using SAS 9.4 (SAS Institute, Cary, NC, USA).

## 3. Results

### 3.1. Participation

Most of the participants who were invited to this study filled out the study survey and questionnaires at a participation rate of >95%.

### 3.2. Participant Characteristics

Participants’ clinical and epidemiological characteristics are summarized in Table 1 for the total population and cases and controls separately. More than half of the participants were males and above the age of 50. Approximately one-third of participants reported being former or current smokers. More than three-quarters had no personal history of IBD nor a family history of CRC. The majority of the participants were either overweight or obese, and more than half had poor physical activity. In total, 43% of the participants had less than 17 years of education. In regard to dietary intake, the majority of participants consumed <18 g/d fibers, compared to USDA recommendation of 25–38 g/d for adults up to age 50 and 21–30 g/d for adults >50 years, and more than half consumed <4 servings of FV per day compared to the USDA recommendation of consuming 5–13 servings. Cases and controls were significantly different in terms of age, gender, smoking status, personal history of IBD, family history of CRC, and diabetes. 

Table 2 represents selected epidemiological characteristics according to fiber and FV intake categories. Participants in the lowest categories of fiber and FV intakes tended to be over 50 years, when CRC risk is increased [19], overweight or obese, have a personal history of IBD, and a family history of CRC. In contrast, participants in the highest categories of fiber and FV intakes were more physically active. Current and former smokers and participants with high education levels tended to be in the highest category of fiber and in the lowest category of FV intake.

### 3.3. Fibers and CRC Risk

Table 3 shows the OR and 95% CI of fiber intake and CRC risk in the three models. In Model 1, adjusting for demographics (age and sex only), increasing fiber intake had a significant inverse association with CRC risk. However, when we added clinical variables (IBD and family history of CRC) in Model 2, the association between fiber intake and CRC risk was no longer significant. However, after adjusting for demographics (age and sex), clinical (IBD and family history of CRC), and behavioral (educational level, PA, smoking status, and BMI) variables in Model 3, an increased fiber intake (Q4) had a significant inverse association with CRC risk.

### 3.4. FV and CRC Risk

Table 4 shows the OR and 95% CI of FV intake and CRC risk in the three models. In Model 1, higher FV (Q4) intake was significantly associated with reduced CRC risk. Results were no longer statistically significant after adjusting for behavioral and clinical covariates in Models 2 and 3.

## 4. Discussion

Findings from this study suggest that higher fiber intake had an inverse association with CRC risk across quartiles of intake, even after adjusting for confounding demographic, behavioral, and clinical factors. Although there was a significant association between higher FV intake and reduced CRC risk after adjusting for age and sex, the associations diminished after adjusting for behavioral and clinical confounding factors. 

The World Cancer Research Fund/American Institute for Cancer Research (WCRF/AICR) 2018 report concluded consuming 90 g of wholegrains per day decreases CRC risk significantly by 17% [20], and that consuming at least 400 g of FV and 30 g per day of fiber from food sources lowers the risk [21]. 

The results of this study are consistent with findings from a recent systematic review and meta-analysis study that included 21 primary observational studies, which found that high fiber intake resulted in a 30% reduced risk of colorectal adenoma with a borderline significant negative linear correlation between the amount of fiber and colorectal adenoma risk [22]. Another large cohort study (NIH-AARP Diet and Health Study) examined the association between fibers and the risk of CRC among 478,994 US adults, and followed them up over 16 years, showing that there was an inverse association of wholegrains intake with CRC incidence, but not fibers [23]. Furthermore, fiber from grains, but not other sources, was associated with a lower incidence of CRC. 

It is suggested that fibers reduce CRC risk due to several beneficial mechanisms in the gastrointestinal site. These include decreasing bile reabsorption, enhancing fecal carcinogens excretion by binding to them, and promoting the gastrointestinal microbiota, specifically short-chain fatty acid (SCFA), which have anti-proliferative effects and can promote gastrointestinal microbiota growth [24,25,26,27]. Wholegrains are also rich sources of anti-carcinogenic compounds, such as vitamin E, selenium, copper, zinc, lignin, phytoestrogens, and phenolic compounds [28]. Another mechanism is that fiber prolongs gastric emptying, thereby reducing the rate of glucose absorption and plasma insulin levels, consequently improving insulin sensitivity and regulating hormones, which stimulate postprandial insulin release, enhance glucose tolerance, and delay gastric emptying [29,30], which prevents obesity [31], one significant risk factor of CRC.

Our study suggests that intake of >5 servings/d of FV may be associated with a 34% reduction in CRC risk when considering age and sex as confounders; however, this association did not remain when we included behavioral confounders (educational level, PA, smoking status, and BMI) and clinical confounders (IBD and family history of CRC). The reason for this null association is unclear. It could be due to chance or study limitations in the design. Results from Model 1, the demographics only model, were consistent with a prospective cohort study of 61,463 Swedish women who were followed up during ~9.6 years [32]. Results revealed that after adjusting for caloric intake, age, and intakes of red and processed meats, total FV consumption was inversely associated with CRC risk, primarily due to fruit consumption. Individuals who consumed >1.5 servings of FV per day had a significant 65% higher chance for developing CRC compared to individuals who consumed >2.5 servings. 

Our results were consistent with a prospective cohort study by the European Prospective Investigation into Cancer and Nutrition (EPIC) that included >500,000 participants from 10 European countries to examine the associations between FV and fiber consumption and the risk of cancer [33]. Results from that study revealed that the risk of CRC was a significant inverse association of 14% with intakes of total FV and of 17% with intake of total fiber. Additionally, the EPIC study revealed that concerning fiber type, cereal fiber intake was significantly inversely associated with CRC risk by 13%, but there was a null association with intakes of fruit or vegetable fiber. Similarly, a recent meta-analysis of 10 prospective studies concluded that the dose–response curve for the association between vegetable and fruit fibers and colorectal cancer risk was non-linear [34].

The consumption of FV and fibers among Palestinians has decreased in recent years. The WHO national population-based most recent survey of NCD risk factors (Palestinian STEPS survey) 2010–2011 showed that the mean number of servings of FV consumed on an average day was 1.0 (1.1 for males and 1.0 for females) for fruits and 1.8 (1.8 for males and 1.7 for females) for vegetables [13]. This is considered low compared to the WHO/FAO recommendation of consuming >5 servings/d of FV [35]. Additionally, 86% (85.5% males and 86.5% females) consumed <5 servings of FV on an average day [13]. A survey conducted in 2014 by the Applied Research Institute Jerusalem (ARIJ) showed that the average annual Palestinian household consumption distribution over the agro-commodities was 42% to field crops (wheat, potatoes, and onions), 40% to vegetables, and 14% to fruit trees [36]. Due to deficiency in cultivated areas, high water prices, high prices of agriculture inputs, and the Israeli occupation policy, most households depend on purchasing their agro-commodities from the FV markets. However, the prices of FV are increasing; fruits increased by 15.6% and vegetable prices increased by 14.4% [36].

People living in the Palestinian territory have high levels of poverty, unemployment, and low educational levels. Since FV are expensive in the Palestinian territories, intakes highly depend on socioeconomic status (SES). Furthermore, we have noticed that intakes of FV and fibers vary by geographic area. For example, people who live in East Jerusalem, under Israeli sovereignty, are more educated and have higher SES than people living in the West Bank and Gaza Strip. Additionally, we have noticed that intakes of FV and fiber were higher among healthy controls who live in East Jerusalem compared to cases and controls who live in the West Bank or Gaza Strip.

Our study’s limitations include a lack of data collection on other dietary components that increase CRC risk, such as caloric intake and consumption of red and processed meats. It also lacks clinical information on specific CRC sites, stages, diagnosis, and treatment. When we assessed PA, we did not use a validated questionnaire such as IPAQ to keep it simple for the participants, as PA was not our main outcome. IBD, a known CRC risk factor, is underdiagnosed among Palestinians, which might cause type II errors. Another limitation is that the screener we used was designed for the American population. There is no FFQ validated to assess FV and fiber intake among Palestinians. However, investigators of this study are familiar with foods that are common among Palestinians and verified the use of this screener for this study. Moreover, the screener did not have the specificity necessary to conduct sub-analyses on specific types of fiber or FVs consumed (e.g., insoluble fiber, soluble fiber, wholegrains, green leafy vegetables, citrus fruits, etc.) nor could assess caloric intake, which we could not adjust for when looking at the association between fiber and FV intake and CRC risk. There is also a possibility for recall bias due to a possibility that cases had modified their eating habits after their diagnosis with CRC. The study was limited to the potential selection bias of controls. One part of the healthy controls group was selected from the same community as the cases, where we included the cases’ companions who lived in the same household, but we did not undertake tests to ensure that the companions who were at a risk of CRC due to family history or due to the same eating habits, were actually disease free. The second part was diabetic patients at risk of developing CRC and were given nutritional counseling on healthy eating habits. Additionally, the third part was a random sample from the community. Another limitation was that our data lacked demographic information on the participants’ regions (East Jerusalem, the West Bank, and Gaza Strip). Additionally, it is known that one of the CRC risk factors is alcohol consumption. Still, we did not include any questions about alcohol consumption due to cultural sensitivity, knowing that alcohol consumption is not common among Muslim Palestinians, and many people may be offended by such a question. 

Our study’s primary strength is that, to our knowledge, it is the first study that has collected dietary information and health-related factors from Palestinian patients diagnosed with CRC. Another strength includes the representative sample of the CRC patients at AVH of the Palestinian population, where patients come from all over the West Bank and Gaza Strip to receive cancer treatment at AVH because some cancer treatments are not available elsewhere. Additionally, results from this study will allow us to carry out prevention policies to prevent the Palestinian community from changing their eating habits from being traditional and Mediterranean to more Western and carbohydrate-rich, that will potentially affect their health.

## 5. Conclusions

In conclusion, high fiber intake was inversely associated with CRC risk among a Palestinian sample when we considered demographic (age and sex) and behavioral (educational level, PA, smoking status, and BMI) confounders. High intake of FV was inversely associated with CRC risk only when we considered age and sex as possible confounders. Our study suggests that a high intake of fiber (18–39 g/d), which can be found in fruits, vegetables, and wholegrains, may reduce the risk of developing CRC. Developing and testing culturally relevant dietary interventions to promote increasing fiber intake among Palestinians is needed, particularly interventions targeting those at high risk for developing CRC. Studies with strong designs are necessary to clarify the effect of FV intake on CRC risk.

## Figures and Tables

**Table 1 ijerph-19-07181-t001:** Descriptive statistics for participants’ clinical and epidemiological characteristics (N = 528).

Variable	Characteristics	Total Study Population (%)	Cases (%) N = 118	Controls (%) N = 410	*p*-Value
**Sex**		527 ^4^	118	409	
	Female	304 (57.7)	55 (46.6)	249 (60.9)	0.006 ^7^
	Male	223 (42.3)	63 (53.4)	160 (39.1)	
**Age (years)**		528	118	410	
	<40	159 (30.1)	11 (9.3)	148 (36.1)	
	40–49	93 (17.6)	32 (27.1)	61 (14.9)	<0.0001 ^8^
	50–59	148 (28.0)	37 (31.4)	111 (27.1)	
	60–69	101 (19.1)	25 (21.2)	76 (18.5)	
	70+	27 (5.1)	13 (11.0)	14 (3.4)	
**Smoking status**		527 ^4^	117	410	
	Never	381 (72.3)	75 (64.1)	306 (74.6)	
	Former	46 (8.7)	16 (13.7)	30 (7.3)	0.04 ^8^
	Current	100 (19.0)	26 (22.2)	74 (18.0)	
**IBD**		527 ^4^	117	410	
	Yes	60 (11.4)	32 (27.4)	28 (6.8)	<0.0001 ^7^
	No	467 (88.6)	85 (72.6)	382 (93.2)	
**Family history of CRC**		528	118	410	
	Yes	36 (6.8)	21 (17.8)	15 (3.7)	<0.0001 ^7^
	No	492 (93.2)	97 (82.2)	395 (96.3)	
**BMI (kg/m^2^)**		505 ^5^	115	390	
	Underweight	4 (0.8)	0 (0)	4 (1.0)	
	Normal	150 (29.7)	33 (28.7)	117 (30.0)	0.56 ^8^
	Overweight	161 (31.9)	41 (35.7)	120 (30.8)	
	Obese	190 (37.6)	41 (35.7)	149 (38.2)	
**Years of education**		517 ^6^	113	404	
	≤14	137 (26.5)	33 (29.2)	104 (25.7)	0.11 ^8^
	15–16	87 (16.8)	25 (22.1)	62 (15.3)	
	≥17	293 (56.7)	55 (48.7)	238 (58.9)	
**Type II diabetes**		528	118	410	
	Yes	227 (43.0)	24 (20.3)	203 (49.5)	<0.0001 ^7^
	No	301 (57.0)	94 (79.7)	207 (50.5)	
**Physical activity (PA)**		528	118	410	
	Poor	286 (54.2)	66 (55.9)	220 (53.7)	
	Moderate	208 (39.4)	45 (38.1)	163 (39.8)	0.90 ^8^
	Good	34 (6.4)	7 (5.9)	27 (6.6)	
**Fruit and vegetable (FV) intake ^1^**		527 ^4^	118	409	
	Very low (0–2 servings/d)	216 (41.0)	43 (36.4)	173 (42.3)	
	Low (3 servings/d)	88 (16.7)	21 (17.8)	67 (16.4)	0.73 ^8^
	Medium (4–5 servings/d)	104 (19.7)	25 (21.2)	79 (19.3)	
	Adequate (>5 servings/d)	119 (22.6)	29 (24.6)	90 (22.0)	
**Fiber intake**		527 ^4^	118	409	
	Q1 (6–12 g/d)	147 (27.9)	36 (30.5)	111 (27.1)	
	Q2 (13–14 g/d)	102 (19.4)	23 (19.5)	79 (19.3)	0.41 ^8^
	Q3 (15–17 g/d)	140 (26.6)	35 (29.7)	105 (25.7)	
	Q4 (18–39)	138 (26.2)	24 (20.3)	114 (27.9)	
**Group**		528			
	Cases ^2^: Diagnosed with CRC	118 (22.3)			
	Controls ^3^: Not diagnosed with CRC	410 (77.7)			

^1^ 1-serving of fruit equals 1 cup of raw, cooked fruit; 1 cup of 100% fruit juice; or ½ cup dried fruit and 1-serving of vegetables equals 1 cup of raw or cooked vegetables; 1 cup vegetable juice; or 2 cups of leafy greens; ^2^ patients with CRC of whom 23 (4.4%) were diagnosed with type II diabetes; ^3^ healthy individuals, of whom 202 (49.3%) were diagnosed with diabetes mellitus; ^4^ has one missing datum; ^5^ has 23 missing data; ^6^ has 11 missing data; ^7^ Fischer’s exact two-sided test; ^8^ chi-square two-sided test; BMI: body mass index; CRC: colorectal cancer; FV: fruits and vegetables intake; IBD: irritable bowel disease.

**Table 2 ijerph-19-07181-t002:** Selected epidemiological characteristics according to fiber and FV quartiles.

	Age (>50), y ^1^	Male %	Current/ Former Smoker %	IBD %	Family History of CRC %	BMI (Overweight and Obese %)	Education (>17 Years) %	PA (Good) %
**Fiber intake**	N = 276 ^1^	N = 223 ^2^	N = 146	N = 60	N = 36	N = 351 ^2^	N = 293	N = 34 ^1^
Q1	33.09	3.60	5.48	33.33	44.44	30.86	19.11	6.06
Q2	22.55	11.26	12.33	13.33	13.89	20.00	16.04	6.06
Q3	24.00	38.29	40.41	30.00	16.67	26.00	29.01	36.36
Q4	20.36	46.85	41.78	23.33	25.00	23.14	35.84	51.52
***p*-value**	0.016 *	<0.0001 *	<0.0001 *	0.13	0.04 *	0.03 *	<0.0001 *	<0.0001 *
**FV intake**	N = 276 ^2^	N = 223 ^2^	N = 146	N = 60	N = 36	N = 351 ^2^	N = 293	N = 34 ^1^
Q1	37.82	38.29	41.10	41.67	52.78	42.57	38.57	21.21
Q2	18.18	15.32	18.49	13.33	11.11	15.43	15.02	21.21
Q3	18.55	23.42	16.44	20.00	13.89	18.00	23.21	27.27
Q4	25.45	22.97	23.97	25.00	22.22	24.00	23.21	30.30
***p*-value**	<0.0001 *	<0.0001 *	<0.0001 *	0.01 *	0.001 *	<0.0001 *	<0.0001 *	0.85

^1^ Age > 50 is considered at a high risk for developing CRC; ^2^ has one missing datum; * indicated significance at *p* < 0.05; BMI: body mass index; FV: fruits and vegetables intake; IBD: irritable bowel disease; PA: physical activity.

**Table 3 ijerph-19-07181-t003:** Odds ratios and 95% confidence intervals between fiber intake and colorectal cancer risk (N = 528; cases: (N = 118), controls (N = 410)).

Model 1 ^1^	OR	95% CI
Q 1	1.00	-
Q 2	0.72	0.38–1.36
Q 3	0.65	0.33–1.28
Q 4	0.40	0.19–0.85
*p*-value	0.12	
*p*-trend	0.02 *	
**Model 2 ^2^**	**OR**	**95% CI**
Q 1	1.00	-
Q 2	0. 92	0.47–1.81
Q 3	0.79	0.39–1.63
Q 4	0.48	0.22–1.06
*p*-value	0.25	
*p*-trend	0.07	
**Model 3 ^3^**	**OR**	**95% CI**
Q 1	1.00	-
Q 2	0.99	0.49–2.01
Q 3	0.76	0.36–1.61
Q 4	0.36	0.15–0.86
*p*-value	0.07	
*p*-trend	0.02 *	

* Indicates significance at *p* < 0.05; ^1^ adjusted for age and sex; ^2^ adjusted for age, sex, IBD, family history of CRC; ^3^ adjusted for age, sex, educational level, PA, smoking status, BMI, IBD, and family history of CRC; CRC: colorectal cancer; CI: confidence interval; OR: odds ratio.

**Table 4 ijerph-19-07181-t004:** Odds ratios and 95% confidence intervals between FV intake and CRC risk (N = 526; cases: (N = 118), controls (N = 410)).

Model 1 ^1^	OR	95% CI
Q 1	1.00	-
Q 2	0.72	0.38–1.36
Q 3	0.66	0.33–1.30
Q 4	0.34	0.15–0.75
*p*-value	0.06	
*p*-trend	0.009 *	
**Model 2 ^2^**	**OR**	**95% CI**
Q 1	1.00	-
Q 2	1.49	0.78–2.83
Q 3	1.41	0.7–2.61
Q 4	1.30	0.72–2.33
*p*-value	0.56	
*p*-trend	0.20	
**Model 3 ^3^**	**OR**	**95% CI**
Q 1	1.00	-
Q 2	1.47	0.75–2.92
Q 3	1.52	0.80–2.91
Q 4	1.18	0.62–2.25
*p*-value	0.55	
*p*-Trend	0.15	

* Indicated significance at *p* < 0.05; ^1^ adjusted for age and sex; ^2^ adjusted for age, sex, IBD, and family history of CRC; ^3^ adjusted for age, sex, educational level, PA, smoking status, BMI, IBD, and family history of CRC; FV: fruits and vegetables; CRC: colorectal cancer; CI: confidence interval; OR: odds ratio.

## Data Availability

De-identified data will be available upon direct contact with the corresponding author.

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
