# Peer review of "A Case–Control Study Examining the Association of Fiber, Fruit, and Vegetable Intake and the Risk of Colorectal Cancer in a Palestinian Population"

_ijerph, 2022, doi:10.3390/ijerph19127181_

Round 1
Reviewer 1 Report
Thank you for the opportunity to review this interesting paper. The authors investigated the very important topic regarding the association between fibers, fruit and vegetables consumption and colorectal cancer risk. With their study the authors provided interesting new data for an undrexplored community such as Palestinian population.
Please find below my comments/questions:
1) Have physical activity level been assessed with any validated tool (for instance IPAQ)?
2) Please indicate the unit of measure for BMI (kg/m2)
3) Pag 3, line 138: Please, check the overweight range (24.9 < BMI < 29.9). The valid range is 25.0 < BMI <29.9 kg/m2)
4) Please, could the authors explain how the sample size has been calculated?
5) I suggest the authors to cite this two systematic reviews (https://doi.org/10.1017/s0007114519001454; https://doi.org/10.3390/ijerph18084168) to support their discussion with a more recent evidence.
Author Response
Response to Reviewer 1 Comments
Point 1: Have physical activity level been assessed with any validated tool (for instance IPAQ)?
Response 1: No. As mentioned on lines 99-102: “PA was assessed by asking if the participant exercises for about 30 minutes for at least 3-4 days/week and what PA is involved in their work (sitting, standing/walking, physical effort, housework, or if they are unemployment)”
We added that to the limitations section “When we assessed PA, we did not use a validated questionnaire such as IPAQ to keep it simple for the participants, as PA was not our main outcome” on lines 285-286.
Point 2: Please indicate the unit of measure for BMI (kg/m2).
Response 2: Added on line 140 and in table 1.
Point 3: Pag 3, line 138: Please, check the overweight range (24.9 < BMI < 29.9). The valid range is 25.0 < BMI <29.9 kg/m2)
Response 3: Edited accordingly.
Point 4: Please, could the authors explain how the sample size has been calculated?
Response 4: This was a convenience sample of 118 cases and 410 controls and no power calculations were performed. Edits are done on lines 74 – 77.
Point 5: I suggest the authors to cite these two systematic reviews (https://doi.org/10.1017/s0007114519001454; https://doi.org/10.3390/ijerph18084168) to support their discussion with a more recent evidence.
Response 5: Information and citations were added to lines 218-222 and lines 257-259, respectively.
Reviewer 2 Report
This case-control study explores the association between fibre intake and the risk of CRC in the Palestinian population. This is a relevant area in the study CRC in an under-studied population.
The strength of this study is in its novelty in the context of under-studied populations and the consideration of cases. The study while designed well can benefit from a more detailed and in-depth description. There is good consideration of the evidence-base in the discussion. However, statements can be better substantiated and contextualised, providing further justification of chosen methods. Results presented in tabular format need to be briefly discussed in-text within the result section.
This study provides an important insight into the importance of socio-economic status, and fruit and vegetable consumption in the development of CRC. It would benefit this manuscript to discuss implications in more depth and consider the implications for practice.
Further comments relating to specific sections can be found below:
- Lines 59 - 61 Statements need to be substantiated
"These include obesity, lack of regular 59 physical activity (PA), smoking, alcohol drinking, and eating patterns high in red or processed meat and low in fruits, vegetables, and fibers"
- Lines 63 - 65 Statements need to be substantiated
- Lines 93 - 94 Further information is needed on the questionnaire
-Line 126 and 135 Further information needed on categorisation of fibre and PA (e.g. definition of 1 serving / day)
-Lines 206 - 208 Clarity is needed
Author Response
Response to Reviewer 2 Comments
Point 1: Lines 59 - 61 Statements need to be substantiated
"These include obesity, lack of regular 59 physical activity (PA), smoking, alcohol drinking, and eating patterns high in red or processed meat and low in fruits, vegetables, and fibers"
Response 1: Added a citation on line 61.
Point 2: Lines 63 - 65 Statements need to be substantiated
Response 2: Edited accordingly.
Point 3: Lines 93 - 94 Further information is needed on the questionnaire
Response 3: More information on the questionnaire was given on lines 95 – 121.
Point 4: Line 126 and 135 Further information needed on categorisation of fibre and PA (e.g. definition of 1 serving / day)
Response 4: Definition of serving of FV/day is added to the footnotes of table 1. PA categorization definition is added on lines 136-138.
Point 5: Lines 206 - 208 Clarity is needed.
Response 5: Edited on lines 106-108.
Reviewer 3 Report
This is an excellent paper. It is clearly written on the whole in an easy-to-read style. The research is original and important to further research in the population concerned. The content and reference to foregoing work is informative and interesting. The results are clearly set out and the discussion is relevant and concise.
There are a few minor improvements in the text to be considered, but otherwise it is good enough to be published as is.
The suggestions are:
1. In the last paragraph of the introduction an up-to-date reference on risk factors would be helpful.
2. 3rd to last sentence of introduction insert "been" so it reads "---yet been examined ---"
3. 6th line of section 3.2 change poorly" to "poor".
4. Foot notes to Tables 1 and 2 : Should be "One missing datum" not "data".
5. Discussion paragraph 4 sentence 2. Change "production" to "producing". Also the sentence sounds ambiguous - it is the SCFA, not the SCFA-producing species that have the effects. Please modify.
6. Paragraph 7. Line 1; "recent" may be better than "the last".
Sentence 4: Start with "A survey ---" rather than "Findings from ---"
Second to last line of Acknowledgments: remove "and " from "--- and of the manuscript.".
Author Response
Response to Reviewer 3 Comments
Point 1: In the last paragraph of the introduction an up-to-date reference on risk factors would be helpful.
Response 1: Added a reference from the most updated information on the American Cancer Society website (10).
Point 2: 3rd to last sentence of introduction insert "been" so it reads "---yet been examined ---".
Response 2: Edited accordingly.
Point 3: 6th line of section 3.2 change poorly" to "poor".
Response 3: Edited accordingly.
Point 4: Foot notes to Tables 1 and 2 : Should be "One missing datum" not "data".
Response 4: Edited accordingly.
Point 5: Discussion paragraph 4 sentence 2. Change "production" to "producing". Also the sentence sounds ambiguous - it is the SCFA, not the SCFA-producing species that have the effects. Please modify.
Response 5: Edited accordingly.
Point 6: Paragraph 7. Line 1; "recent" may be better than "the last".
Sentence 4: Start with "A survey ---" rather than "Findings from ---"
Second to last line of Acknowledgments: remove "and " from "--- and of the manuscript.".
Response 6: Edited accordingly.
Reviewer 4 Report
My suggestions to the authors is that they give continuity to the study and that its results allow them to carry out prevention policies to prevent the Palestinian community from changing their eating habits that affect their health.
Author Response
Response to Reviewer 4 Comments
Point 1: My suggestions to the authors is that they give continuity to the study and that its results allow them to carry out prevention policies to prevent the Palestinian community from changing their eating habits that affect their health.
Response 1: Added that as the last sentence of the discussion section.